# Supramolecular engineering of charge transfer in wide bandgap organic semiconductors with enhanced visible-to-NIR photoresponse

Yifan Yao [1], Qi Ou [2], Kuidong Wang [1], Haijun Peng [1], Feier Fang[3], Yumeng Shi [3], Ye Wang[1], Daniel Iglesias Asperilla [1], Zhigang Shuai [2] & Paolo Samorì [1✉]

Organic photodetectors displaying efficient photoelectric response in the near-infrared are typically based on narrow bandgap active materials. Unfortunately, the latter require complex molecular design to ensure sufficient light absorption in the near-infrared region. Here, we show a method combining an unconventional device architecture and ad-hoc supramolecular self-assembly to trigger the emergence of opto-electronic properties yielding to remarkably high near-infrared response using a wide bandgap material as active component. Our optimized vertical phototransistors comprising a network of supramolecular nanowires of N,N′-dioctyl-3,4,9,10-perylenedicarboximide sandwiched between a monolayer graphene bottom-contact and Au nanomesh scaffold top-electrode exhibit ultrasensitive light response to monochromatic light from visible to near-infrared range, with photoresponsivity of $2 \times 10^5$ A/W and $1 \times 10^2$ A/W, at 570 nm and 940 nm, respectively, hence outperforming devices based on narrow bandgap materials. Moreover, these devices also operate as highly sensitive photoplethysmography tool for health monitoring.

[1] University of Strasbourg, CNRS, ISIS UMR 7006, Strasbourg, France. [2] Department of Chemistry, Tsinghua University, Beijing, China. [3] Institute of Microscale Optoelectronics, Shenzhen University, Shenzhen, China. ✉email: samori@unistra.fr

Near-infrared organic photodetectors (OPDs), capable to respond to illuminations at wavelengths exceeding 780 nm, have attracted a notable interest during the last few years because of their potential application as motion detector, remote control, night vision, biomedical imaging as well as health monitoring[1–4]. The active molecular material to be used for photosensing, in order to excel in its photoresponsivity, should combine high light absorption in the chosen spectral region to guarantee efficient exciton formation, optimal charge separation, and transport to the respective electrodes. Such characteristics can be found in ordered organic architectures such as aligned films or crystalline low-dimensional nanostructures, whose optoelectronic properties can be tuned by rational molecular design and controlled self-assembly, thereby making π-conjugated organic nanostructures promising candidates for the next generation NIR photodetectors[5–10]. Self-assembled semiconducting organic nanowires are space confined nanostructures with a typical cross-section in the 2–200 nm range and a length ranging from hundreds of nanometers to millimeters, enabling electrons, holes, or photons to travel freely along one dimension, with exciton transport up to a few micrometers[11,12]. Their high aspect ratio, low number of grain boundaries and efficient exciton generation of these subwavelength semiconducting nanostructures make them ideal building blocks for applications in nanoelectronics and nanophotonics to bridge the mesoscopic and macroscopic world[13–15].

Hitherto, various kinds of self-assembled supramolecular nanostructures have demonstrated promising light-response from UV to visible light when embedded as active layers in photodetectors[16–19]. However, to the best of our knowledge, NIR organic phototransistors based on wide bandgap crystalline supramolecular nanowires have not yet been reported[3,20,21]. The wiring-up of discrete and anisotropic nanostructures/microstructures, to ensure a reproducible performance and integration in real supramolecular optoelectronics devices, still represents a major challenge[22–25]. More importantly, wide bandgap organic semiconducting materials are usually considered being non sensitive to infrared light. According to traditional views, the narrow bandgap is the prerequisite for generating light-induced excitons in the NIR region with good photogeneration yield and an excellent NIR response. Because of this reason, photodetectors operating in the NIR region based on organic semiconducting molecular materials are scarce.

Here, we show a vertical phototransistor (VPT) based on a supramolecularly engineered crystalline perylene nanowires network integrated in an ad-hoc junction consisting of a top nanomesh scaffold electrode, exhibiting photoresponse under monochromatic light in a wide range of wavelengths, spanning from the visible to the NIR. In particular, our photonic device shows a high photoresponsivity ($2 \times 10^5$ A/W), fast photoresponse (~10 ms) and high detectivity ($10^6$ Jones) resulting from the efficient charge, exciton transport. Such key performance indicators could be achieved by integrating ca. 900 nm thick film comprising entangled crystalline supramolecular nanowires forming a continuous network in an asymmetric junction consisting of a patterned graphene bottom electrode with a soft-contact gold nanomesh scaffold as top electrode. The markedly high light response in the NIR resulted from the joint effect of the efficient intermolecular charge transfer through the supramolecular nanowires and the unique device design. Our approach represents a powerful alternative the use of complex narrow bandgap organic materials to attain efficient NIR detection, and therefore in may represent a major step forward for the next-generation highly sensitive NIR photodetectors.

## Results

### Preparation of PTCDI-C8 nanowires.
High quality N,N′-dioctyl-3,4,9,10-perylenedicarboximide (PTCDI-C8) n-type crystalline nanowires can be self-assembled by making use of either solvent phase-transfer (PT) or solvent-induced precipitation (SIP). Both these two processing strategies enable to achieve a dynamic balance between the orthogonal good and poor solvent. Significantly, the properties of the formed nanowires including the degree of crystallinity, size, and softness could be further tuned by modifying concentration, temperature, and solvent type[26]. PTCDI-C8 nanowires prepared by phase transfer method are more crystalline and exhibit red shifted resonance absorption peaks, however the high rigidity and large size (both in length and cross section) of PT-nanowires make them not ideal for the generation of homogeneous supramolecular films[27]. Because of these downsides, we decided to exploit solvent-induced-precipitation to self-assemble the PTCDI-C8 molecules into supramolecular nanowires. Such a crystal growth process triggered by the addition of a bad solvent (i.e., ethanol) minimizes the incorporation or trapping of solvent molecules into the crystals, being an optimal strategy to achieve enhanced opto-electronic properties of the nanostructures[22]. In addition, PTCDI-C8 nanowires prepared by SIP exhibit a high flexibility which render them most suitable for the preparation of a uniform closely-packed 3D nanowires network.

Dynamic light scattering (DLS) analysis made it possible to gain insight into the particle size and distribution. It revealed a homogeneous size distribution of the PTCDI-C8 nanowires in the ethanol solution, with a hydrodynamic diameter of 955.4 nm and polydispersity index of 0.31 (Supplementary Figs. 1 and 2). Subsequently, uniform and dry nanowires networks with different thickness were produced by vacuum-assisted stamp method (see "Methods" section). Compared with spin-coating or drop-casting, the use of vacuum-assisted stamp method displays three advantages: (1) densely stacked supramolecular nanowires network can be evenly distributed on any surfaces; (2) all of the material can be fully deposited at surfaces, without any waste; (3) good reproducibility ensuring high device performance. To assess the optoelectronic properties of the chosen organic material, i.e., PTCDI-C8, we have prepared planar phototransistors based on both self-assembled nanowires network and thermally evaporated films in a bottom gate/bottom contact configuration. Compared to PTCDI-C8 evaporated film, the nanowires network shows a broaden tailing peak in the UV-NIR absorption spectrum at 700–1000 nm range (Supplementary Fig. 3). The AFM imaging revealed that the PTCDI-C8 films consist of tightly packed grains with diameters of ca. 200 nm, which are much smaller than the PTCDI-C8 nanowires whose lengths are in the micrometers scale (Supplementary Fig. 2). Although there is no obvious absorption peak in NIR range for PTCDI-C8 nanowire, surprisingly, the opto-electric characterizations illustrated in Supplementary Fig. 4 showed a striking photoresponse to 740, 850, and 940 nm NIR light when the PTCDI-C8 nanowires network are integrated in planar phototransistors with different channel lengths. Interestingly, such response was not observed in similar devices based on thermal evaporated PTCDI-C8 films (Supplementary Fig. 5), suggesting that the order at the supramolecular level plays a key role.

### Quantum mechanics and molecular mechanics simulations of charge transfer within PTCDI-C8 nanowire.
Theoretical calculations have been carried out to explore the NIR photoresponse mechanism of PTCDI-C8. Supplementary Fig. 7a reveals that the $S_1$ state of PTCDI-C8 molecule in solution is dominated solely by a local excitation (LE) transition, giving rise to a small reorganization energy between $S_1$ and the ground state, and narrow absorption spectrum with well-defined vibrational peaks (Supplementary Fig. 7b). The good agreement between the calculated absorption spectrum and one recorded experimentally validates our modeling

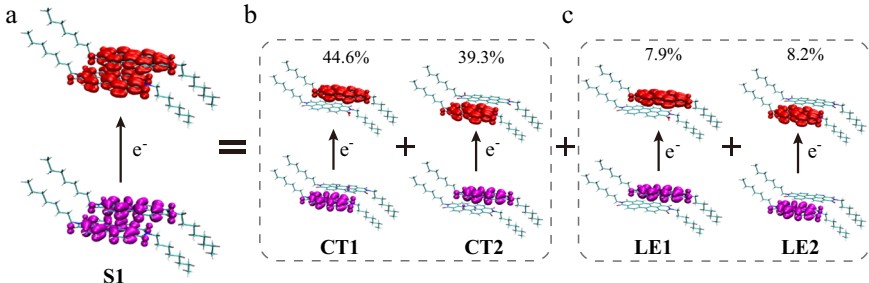

**Fig. 1 Attachment–detachment densities of the S1 state and its composition of PTCDI-C8 dimer inside the crystalline structure. a** Attachment densities are shown in red and detachment densities are shown in magenta. **b**, **c** The four transition components are recognized and quantified via Boys localized diabatization method, intermolecular charge-transfer (**b**); intramolecular charge-transfer (local excitation (LE) transition) (**c**). The two intermolecular charge-transfer (CT) transitions are the dominating of character the S1 state of PTCDI-C8 dimer inside the crystalline structure (**b**), which give rise to the broadened tailing peak in the NIR region.

(Supplementary Fig. 6). To cast light onto the opto-electronic properties of the PTCDI-C8 nanowires, we applied quantum mechanics and molecular mechanics (QM/MM) simulations. We defined a dimer of PTCDI-C8, embedded inside the crystalline structure of the nanowire, as the QM unit (Supplementary Fig. 8)[28]. Figure 1 shows that the $S_1$ state of the PTCDI-C8 dimer comprises four different transitions according to the localized diabatization description[29]. We found that intermolecular charge-transfer (CT) transition between two monomers (CT1 and CT2 in Fig. 1) are dominant with total contributions exceeding 80%, yielding a decrease of the energy of S1 compared to the value in solution (Supplementary Table 1) along with a significant geometrical adjustment upon excitation with larger reorganization energy between S1 and the ground state. Such geometrical change and enlarged reorganization energy significantly broaden the absorption spectrum[30], resulting in an elongated tailing peak in the absorption spectrum at the NIR region (Supplementary Figs. 3 and 6). The absorption tail induced photoresponse of PTCDI-C8 can also be evinced by the fact that the response from 570 to 940 nm is monotonically weakened as shown in Supplementary Fig. 22, which is in good agreement with the monotonic decrease of the photon absorption at wavelength exceeding 570 nm. Altogether, such findings confirms that intermolecular CT determines the photo-response of PTCDI-C8 nanowires in the NIR region.

**Fabrication of vertical field-effect transistor with nanomesh scaffold**. To further improve the efficiency of light-response of devices based on PTCDI-C8 nanowires, a vertical device geometry has been realized. In our vertical field-effect transistors (VFETs) the drain, source electrode, semiconducting layer, dielectric layer, and the gate electrode are vertically stacked[31]. Because of the limited channel length (ca. 900 nm), excitons can be efficiently separated and collected by the source and drain electrodes resulting in more efficiently photoelectric conversion and faster response speed. Figure 2a, b portrays the device fabrication process and layout of the PTCDI-C8 nanowires based VFETs. Nanosphere lithography, via the formation of a hexagonal close-packed (hcp) nanosphere monolayer, is used as a mask to fabricate the nanomesh scaffold. The fabrication procedure starts with the deposition of polystyrene (PS) nanosphere monolayer (Fig. 2a). An oxygen plasma treatment was then applied to widen the gap between adjacent PS nanospheres. A thin layer of gold (80 nm) was then deposited by high-vacuum thermal evaporation followed by a lift-off step (Supplementary Fig. 9). The patterned nanomesh scaffold was detached from the silicon wafer by etching the sacrificial Al layer and transferred onto the targeted device. Figure 2b displays the CVD monolayer graphene that was transferred onto the SiO2/Si substrate and patterned as the

bottom source electrode using photolithography and O2 plasma etching. Raman spectroscopy revealed two sharp peaks at 1585 cm$^{-1}$ for G peak and 2673 cm$^{-1}$ for 2D peak and a high $I_{2D}/I_G$ ratio of 2.0 confirming the high quality of monolayer graphene on SiO2 along with a smooth surface proven by AFM imaging (Supplementary Fig. 10). Such characteristics are important to ensure a good physical contact between graphene and the active organic layer. PTCDI-C8 supramolecular nanowires networks were then deposited onto patterned graphene by vacuum-assisted stamp method. Finally, the soft-contact top gold nanomesh electrodes were positioned on top of the PTCDI-C8 nanowires network through a water-assistance transfer approach to finalize the device fabrication.

Figure 2c, d shows photographs of the resulting gold nanomesh scaffold on Al/silicon substrate and prepared PTCDI-C8 nanowires based vertical devices, respectively. Figure 2e, f reports the topographical AFM images of PTCDI-C8 nanowires network and the prepared vertical device as displayed in Fig. 2d. The honeycomb-like hole-array nanomesh scaffold exhibits a high degree of periodicity. Figure 2g and Supplementary Fig. 11 displays the scanning electron microscope (SEM) image top view evidencing the full-coverage supramolecular nanowires in close contact with nanomesh scaffold. The latter is enabled by the flexibility and stretchability of gold nanomesh scaffold. Moreover, as shown in Fig. 2h, i and Supplementary Fig. 11d–f, the cross-sectional SEM images further illustrate the constructed vertical heterostructures of SiO2-graphene/nanowires network/nanomesh scaffold revealing a stacked architecture with well-defined and sharp interfaces between each layer without any collapsing, especially between the nanowires network and the nanomesh scaffold.

**Photoelectrical characteristics in nanowires network based vertical devices**. The photoelectrical properties of the PTCDI-C8 nanowires network were explored by fabricating VFETs, with the device geometry and working mechanism illustrated in Supplementary Fig. 12. PTCDI-C8 nanowires networks of different thicknesses were produced by the vacuum-assisted stamp method, then covered with the nanomesh scaffold. As shown in Supplementary Fig. 13, the thinner is the PTCDI-C8 nanowires network (0.4 μm) the rougher is the surface and the larger is the amount of porous defects, overall resulting in a large leakage current between the source and drain electrode and no gate regulation. Supplementary Fig. 14 displays the transfer characteristics of PTCDI-C8 nanowires based VFETs with different channel length. They reveal a good $J_{on}/J_{off}$ ratio upon sweeping the gate voltage ($V_G$) from −60 to 80 V, when the channel length of devices larger than 0.9 μm. In fact, too large channel lengths could lead to reduced current density and

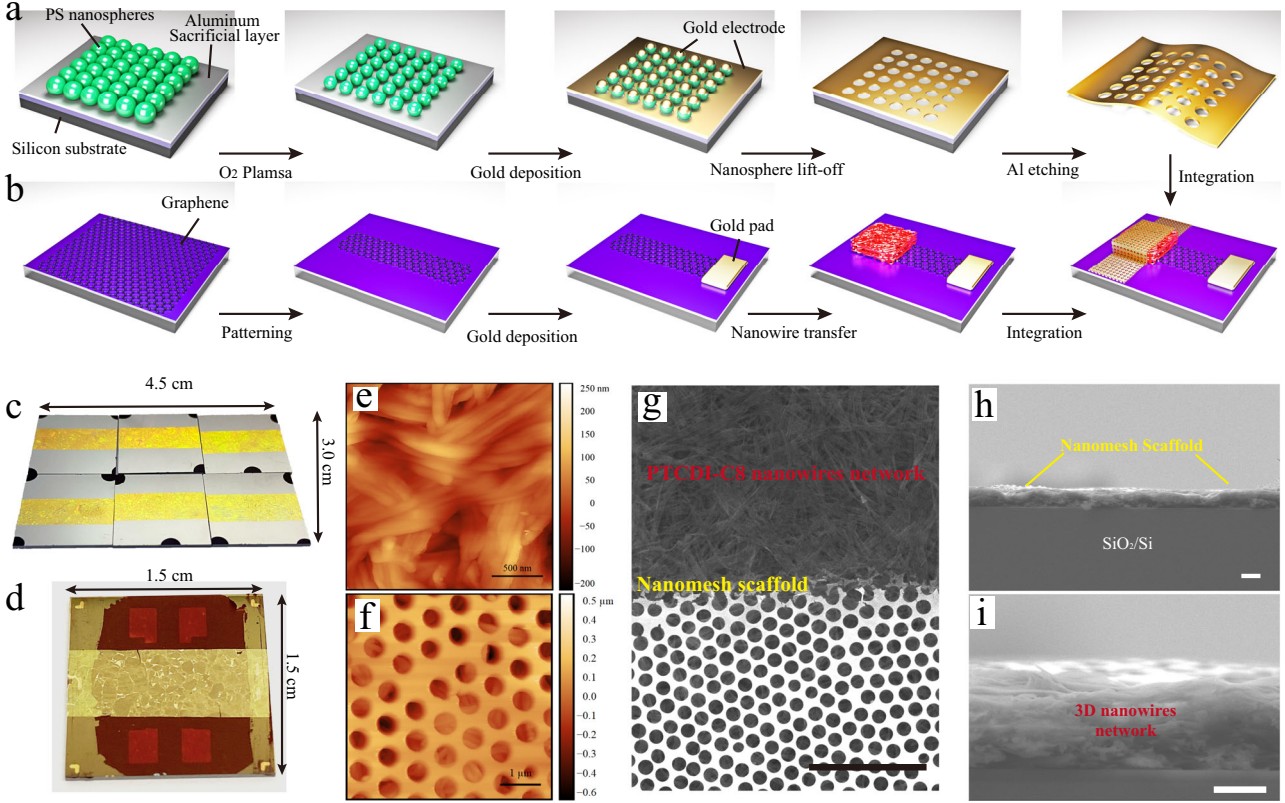

**Fig. 2 Design of a vertical field-effect transistor (VFET) with nanomesh scaffold. a** Fabrication process flow for the hexagonal hole array nanomesh scaffold through nanosphere lithography, $O_2$ plasma etching and transfer process. Nanosphere was transferred onto an Al-deposited $SiO_2$/Si wafer. The geometrical parameters of the resulting nanostructure could be tuned by the diameter of the polystyrene nanospheres and the etching dose of the oxygen plasma. The aluminum layer was removed by solution etching, and the nanomesh scaffold was then transferred onto the targeted substrate. **b** Graphene/nanowires network fabrication procedure. The transferred monolayer graphene was patterned by photolithography and $O_2$ plasma to form the bottom source electrode. PTCDI-C8 nanowires network was transferred on the graphene layer through a vacuum-assisted stamp method. The floating nanomesh scaffold was then transferred directly onto the PTCDI-C8 network as the top drain electrode. **c** Photograph of a large-area nanomesh scaffold on Al-deposited silicon wafers. **d** The resulting PTCDI-C8 nanowires VFETs with nanomesh scaffold. **e** AFM image (topography) of the transferred PTCDI-C8 nanowires network. **f** AFM image of the final vertical-channel PTCDI-C8 device bearing the nanomesh scaffold. **g** SEM image of nanomesh electrodes covering PTCDI-C8 nanowires network, scale bar: 5 μm. **h** Cross-sectional SEM image of a PTCDI-C8 nanowires based vertical device, scale bar: 2 μm. **i** Zoom-in SEM image of the same sample in **h**, the thickness of the PTCDI-C8 nanowires network is around 2 μm, scale bar: 2 μm.

weak gate regulation (Supplementary Fig. 14d, e). In this study, a channel length of 0.9 μm PTCDI-C8 nanowires network was chosen, because of high current density (~0.1 mA/cm²) and $J_{on}/J_{off}$ ratio (~$10^2$). PTCDI-C8 displayed good thermal stability with the thermal decomposition temperature as high as 431 °C (Supplementary Fig. 15). For the sake of comparison, different thermal annealing temperatures have also been studied in a range spanning from 120 to 220 °C. After being annealed at 160 °C for 30 min in the glovebox, the devices showed a balanced current density ($J$) 0.1–0.3 mA/cm² and $J_{on}/J_{off}$ ratio $10^2$–$10^3$ (Supplementary Table 2).

The high performance of PTCDI-C8 nanowires network based VFETs with high $J_{on}/J_{off}$ ratio as well as relatively high current density holds potential for application in integrated optoelectronic devices, such as vertical phototransistors (VPTs), which combine both signal detection and amplification (Supplementary Fig. 16). Figure 3a displays the transfer curves of PTCDI-C8 nanowires VPTs measured in the dark and 570 nm green light with different irradiance. A significant increase of source-drain current can be observed even under very small irradiance (24 nW/cm²). Figure 3b exhibits the 2D color-filling contour image of the photocurrent density along with wavelength (300–690 nm) and irradiance variation, demonstrating the highest photocurrent at a wavelength around 570 nm, which is consistent with its light absorption property (Supplementary Figs. 6 and 16c). Compared

to the dark state depicted in Supplementary Fig. 17a, the output characteristics of the vertical phototransistor at the various gate voltages present a significant increase in the source-drain current under 102 nW/cm², 570 nm green light. In traditional organic phototransistors with planar configuration, the variation of photocurrent is notably weakened in the accumulation region, which means the devices can only operate in the depletion region. However, the photocurrent in the PTCDI-C8 nanowires VPTs shows prominent changes in both the depleted region and the accumulated region. More importantly, when working at the accumulation region, devices exhibit much higher responsivity to the less intense light (~nW/cm²) than at the depletion region. This could be ascribed to the short channel length facilitating the charge transfer. Moreover, when devices work at accumulated region, the field-induced electrons can efficiently fill the traps at graphene/PTCDI-C8 nanowire heterointerface or at the nanowire itself, and as a result, the photo-generated electrons will not be quenched by the traps leading to an improved faint light responsivity[32,33]. Figure 3c exhibits the current density shift of the vertical phototransistor as a function of the irradiance in the log-log plot both at depleted area ($V_G = -50$ V) and accumulated area ($V_G = 80$ V). It reveals a linear relationship with irradiance at a wide range. Linear dynamic range (LDR), is one of the important parameters to evaluate the performance of

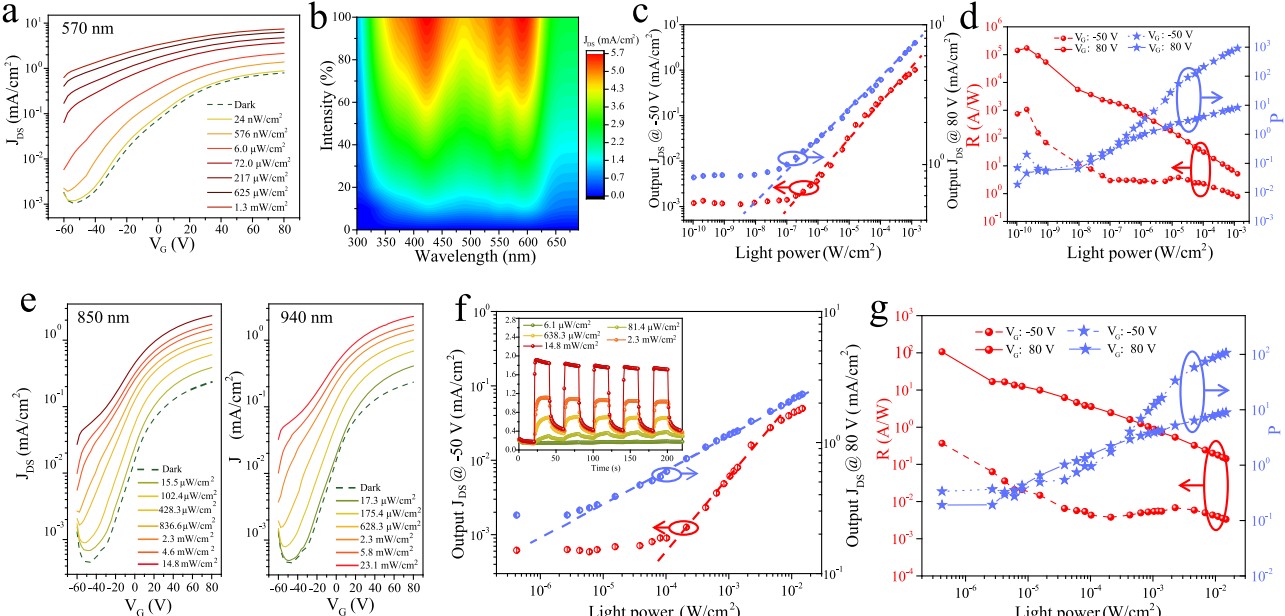

**Fig. 3 Characteristics of PTCDI-C8 nanowires network vertical phototransistors (VPTs). a** Transfer characteristics of phototransistors under 570 nm with various irradiances compared to that in darkness, $V_{DS} = 20$ V. **b**, 2D map of the photocurrent density (J) recorded by scanning the irradiation wavelength and irradiance, at $V_G = -50$ V, $V_{DS} = 20$ V. **c** Photocurrent density value at $V_G = -50$ V (red dots) and $V_G = 80$ V (blue dots) plotted against different irradiances at 570 nm. **d** Photoresponsivity (R) and photosensitivity (P) versus different irradiances in the depleted area ($V_{DS} = 20$ V, $V_G = -50$ V, dash line) and in the accumulated area ($V_{DS} = 20$ V, $V_G = 80$ V, solid line), the photoresponsivity and photosensitivity can reach $\approx 2 \times 10^5$ A/W and $1 \times 10^3$ at 570 nm. **e**, Transfer curves of PTCDI-C8 nanowires VPTs with $V_{DS} = 20$ V under dark condition or different irradiances at 850 and 940 nm. **f** Photocurrent density value at $V_G = -50$ V (red dots) and $V_G = 80$ V (blue dots) plotted against different irradiances at 850 nm. Inset: The photoswitching cycles at 850 nm working at accumulated region driven by $V_{DS} = 20$ V and $V_G = 60$ V. Y-axis demonstrates current density J (mA/cm²). **g** R and P versus different irradiances in the depleted area ($V_{DS} = 20$ V, $V_G = -50$ V, dash line) and in the accumulated area ($V_{DS} = 20$ V, $V_G = 80$ V, solid line), the photoresponsivity and photosensitivity can reach $\approx 1 \times 10^2$ A/W and $1.1 \times 10^2$ at 850 nm.

photodetctors[34]. As shown in Fig. 3c, there is a linear relationship between the logarithm of the current density and the logarithm of the irradiance in the range $10^{-8}$–$10^{-3}$ W/cm² at accumulated area (80 V). The calculated LDR measured by the photoresponse characteristic amounts to 72 and 87 dB in its depleted region and accumulated region, according to LDR $= 20\lg(P_{max}/P_{min})$ (1), where $P_{max}$ ($P_{min}$) is the starting (or ending) point of light intensity when the photocurrent density follows linearly to the light intensity.

Photoresponsivity (R) and photosensitivity (P) are also important parameters to evaluate the performance of phototransistors and are defined as R $= (I_{Light} - I_{Dark})/(S \times P_{In})$ (2) and P $= (I_{Light} - I_{Dark})/I_{Dark}$ (3) with $I_{Light}$ and $I_{Dark}$ being the drain current under illumination and in dark, respectively. $P_{In}$ is the incident light power density (W/cm²), and $S$ is the effective device area (cm²)[8]. As shown in Fig. 3d, the photoresponsivity decreased when increasing the light intensity. According to the Eqs. 2 and 3, remarkable photoresponsivity (R, A/W) and photosensitivity (P) $1.7 \times 10^5$ A/W and $1.3 \times 10^3$ are obtained at 570 nm, respectively. External quantum efficiency (EQE) and specific detectivity ($D^*$, Jones) are important factors in the assessment of the photodetectors performance and the corresponding calculation equations are defined as follows: EQE $= Rhc/q\lambda$ (4) and $D^* = \sqrt{A\Delta f}/NEP$ (5); where $h$ is the Planck's constant, $c$ is the speed of light in a vacuum, $\lambda$ is the incident light wavelength, and $q$ is the elementary charge. For specific detectivity, NEP $= i_{noise}/R_\lambda$ where $A$ is the working area of the device and $\Delta f$ is the bandwidth (0.833 Hz) of the equipment, and $i_{noise}$ is the total noise current (detailed measurement results are reported in Supplementary Fig. 18)[35]. The maximum EQE can be calculated

as $\approx 4 \times 10^7\%$ with a power intensity of 0.2 nW/cm² at the accumulated region. The numerical value of the shot noise is ten times larger than that of the thermal noise, therefore $D^*$ can be expressed as $D^* = R/\sqrt{2qJ_{Dark}}$ (6), where $J_{Dark}$ is the dark current density of the phototransistors. Supplementary Fig. 17b plots $D^*$ of the vertical phototransistors as a function of the incident irradiance. Thanks to the ultra-high photosensitivity of PTCDI-C8 nanowires VPTs, the maximum $D^*$ can reach up to $10^{16}$ Jones. The device shows a very stable dark current after many cycles of successive light on and off. When the device is exposed to the irradiance of 33 mW/cm², the drain current mounted up instantly with the on/off current ratio up to 18 (Supplementary Fig. 17c). We recorded very sharp peaks by scanning the monochromatic light wavelength between 570 and 690 nm rapidly (Supplementary Fig. 17d), a fast time response with rise and decay times of 9.7/28.2 ms was measured, which is far better than the most of single component or heterojunctions organic phototransistors[32,36–38]. This phenomenon strongly suggests that our nanowires-based vertical phototransistor is able to record fast wavelength scanning within milliseconds with appreciable accuracy.

As shown in Fig. 3b, the PTCDI-C8 nanowires VPTs also exhibited a very high photoresponse in the range 650–690 nm, even though there is no obvious absorption peak at these range for PTCDI-C8 nanowires (Supplementary Fig. 6). Our VPTs based on PTCDI-C8 nanowires network have exhibited high NIR response from 740 to 940 nm, largely exceeding the one described above for planar geometry, although PTCDI-C8 is a well know wide bandgap material with the bandgap as large as 2.2 eV[22]. Figure 3e shows the typical transfer curves obtained by varying

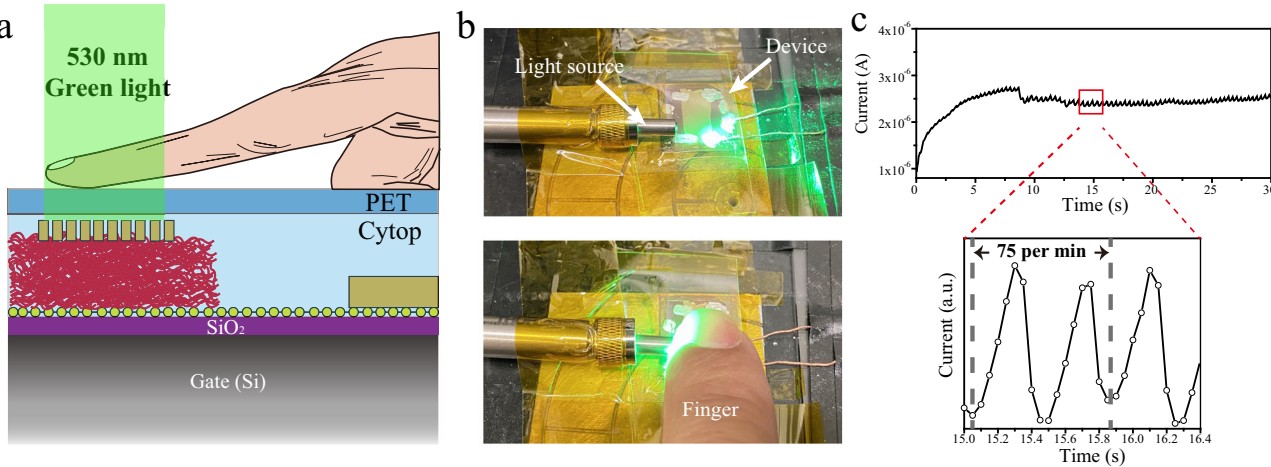

**Fig. 4 Supramolecular nanowires system comprising health-monitoring sensors. a** Device configuration of the transmission-based photoplethysmogram (PPG) sensor. **b** Photographs of the setup for the real-time heart-rate (HR) measurement based on the PTCDI-C8 nanowires vertical phototransistors. **c** Long-term measurement of the pulse signal measured from vertical phototransistors at resting condition.

the NIR light intensity from 15.5 μW/cm² to 23.1 mW/cm². It reveals that the photocurrent shifts towards higher values as the irradiance increases at both depleted region (−50 V) and accumulated region (80 V). Figure 3f exhibits the output current density shift of the vertical phototransistor as a function of the irradiance in the log–log plot under 850 nm. It illustrates a very good linear relationship with light intensity, and the maximum current density is 2.3 mA/cm² at the light intensity of 14.8 mW/cm², at accumulated region. When the device is working at accumulated region, the VPTs show a very good linear relationship at low irradiance (6.1 μW/cm²), not being saturated at high irradiance (14.8 mW/cm²). The calculated LDR is 27 and 68 dB in its depleted region and accumulated region under 850 nm NIR light. Compared with the dark state as shown in Supplementary Fig. 20a, the output characteristics of the transistor at the various gate voltages also exhibit a significant increase in the source-drain current under NIR light. The vertical phototransistor also showed current switching upon irradiation without noticeable fatigue (Fig. 3f inset and Supplementary Fig. 20b). According to the equations 2 and 3, remarkable R and P of $1.1 \times 10^2$ A/W and $1.0 \times 10^2$ are obtained at 850 nm NIR light, respectively (Fig. 3g). The calculated EQE can reach ≈$2 \times 10^4$% at the accumulated region. Supplementary Fig. 20c shows the $D^*$ of the vertical phototransistors as a function of the incident irradiance. In the accumulated region, the calculated maximal $D^*$ can reach ≈$1.2 \times 10^{13}$ Jones under an incident intensity of 0.4 μW/cm², which is superior to some of the reported narrow bandgap organic-based NIR photodetectors[38–41]. When we extended the NIR light to 740 and 940 nm, the PTCDI-C8 nanowires VPTs have also validated excellent NIR light photoresponse, which are similar to the results obtained at 850 nm NIR light, as shown in Supplementary Figs. 19–21. Our devices also demonstrate good reproducibility and stability (see Supplementary Figs. 23 and 24).

As control experiment, we have prepared PTCDI-C8 nanowires VPTs based on a continuous Au film as top electrode, instead of the nanomesh one. After transferring the gold film onto the PTCDI-C8 nanowires network, Supplementary Fig. 25 reveals a corrugated gold film, implying a low-quality contact with the bottom PTCDI-C8 nanowires network. This is due to the fact that continuous gold films lack stretchability compared with the gold nanomesh scaffolds. Not surprisingly, the devices with continuous gold film demonstrated much worse optoelectronic performance compared to the one with nanomesh scaffold, both

in terms of photosensitivity, photoswitching range of current and response time (Supplementary Fig. 26). Such an evidence suggests that the nanomesh scaffold is an indispensable component for achieving high-performance in the vertical phototransistors, due to the better contact with the unflatten PTCDI-C8 nanowires network and higher transparency to the light because of thousands of nanoholes. It should be noted that the gold nanomesh scaffold may form a plasmonic structure that can also potentially act as an antenna to further boost the absorption if properly optimized[42,43], but no clear evidence for this effect could be detected in the present study, probably due to the low-quality factor of the nanomesh scaffold prepared by nanosphere lithography or the rough nanomesh/nanowires interface[22].

**Real-time health monitoring based on supramolecular network.** Our supramolecular nanowires VPTs are remarkably sensitive owing to their highly crystalline nanostructure and specifically designed vertical configuration with nanomesh scaffold. All these qualities are important to boost the device's optoelectronic performance. As a proof-of-principle assessment of the practical application of the supramolecular network-based VPTs, we carried out a photoplethysmography (PPG) test using our VPTs[44,45]. The basic working principle is the light emitted from the LEDs is partially absorbed, reflected, and/or scattered by human tissues, which can be detected by a photodetector. As a result of the change in blood volume upon each cardiac cycle and the pulsatile signal can be extracted to evaluate the heart rate. By combining 530 nm green light with our VPTs, Fig. 4a shows the device structure of the transmission mode PPG measurement. To detect pulse waves, we put a finger over on top of the VPTs, and the green light was emitted from the left side of the finger then pass through the fingertip and the transmission light was detected by the VPTs (Fig. 4b). With the driving voltage of the VPTs set at 20 V, the current of the phototransistor was monitored to measure the absorption of green by the blood. Figure 4c and Supplementary Movie 1 show the current change of VPTs. The PPG signal generates little noise and good repeatability, which is enabled by the ultra-high sensitivity of VPTs to the green light. Our device possesses good stability in air, with the PPG signal remaining constant after a long-time measurement. We succeeded in measuring periodic cycles, although the systolic and diastolic peaks in the PPG profile are slightly deviated from the

normal waveform, because of the charge's accumulation and response speed limitation[46,47]. We can distinctly extract the pulsatile signal from the output current, in which the heart rate was measured to be 75 beats/min in a quiet condition.

## Discussion

In summary, we have reported NIR response from a wide bandgap organic prototypical semiconductor, i.e., PTCDI-C8. Such unique opto-electronic characteristics are the result of an ad-hoc supramolecular self-assembling ensuring efficient intermolecular charge transfer, which determines the emergence of a tailing peak in the absorption spectrum in the NIR region. To further boost the light response both in the visible and NIR region, we have devised a vertical field-effect phototransistor, in which the network of PTCDI-C8 supramolecular nanowires are sandwiched between a CVD graphene bottom-contact and Au nanomesh scaffold top-electrode. Our device exhibits numerous advantages. First, crystalline nanowires with the vertical device configuration feature a fast photoresponse (~10 ms) owing to the reduced source-drain distance. Second, heterojunction of PTCDI-C8 nanowires network and graphene endows excellent electrical and optoelectronic performances. More importantly, we show that by taking advantage of supramolecular self-assembly determining efficient intermolecular charge transfer, enhanced light sensing in the 740–940 nm range can be obtained using wide bandgap semiconductors, with performance surpassing those of some well-established narrow bandgap materials. Third, the device geometry guarantees a current flow only across the vertical nanowires area and minimizes the unintentional influence of interface defects or traps. Our model PTCDI-C8 nanowires VPT shows key performance indicators such as photoresponsivity of $2 \times 10^5$ A/W and $1 \times 10^2$ A/W, detectivity of $10^{16}$ and $10^{13}$ Jones, at 570 nm and 940 nm, respectively. Real-time heart rate monitoring by PPG was demonstrated with the optimized devices, providing clear evidence of the huge potential of the supramolecular based photodetectors for practical implementation as a highly efficient photodetection platform. Our versatile device can be further optimized in the future. For example, stretchable polymer insulators and substrates could be used to endow flexibility and stretchability to the vertical phototransistors, taking advantage of the intrinsic stretchability of components including the graphene electrode, the nanomesh scaffolds, and the PTCDI-C8 networks. In order to improve separation and transport of light induced-excitons, to ultimately attain faster light response and higher NIR responsivity, active layers combining n-type nanowires with p-type supramolecular nanostructures can be employed. Our vertical channel device configuration could be exploited to perform fundamental studies in other nanostructured optoelectronic and photovoltaic devices based on various organic/inorganic materials exhibiting fascinating optical properties, towards the emergence of photonic technologies.

## Methods

**Materials**. The monodispersed suspension of 800 nm PS nanospheres (10wt%, in water) and N,N'-dioctyl-3,4,9,10-perylenedicarboximide (98%) were purchased from Sigma-Aldrich without further purification. The silicon wafer was purchased from Fraunhofer with a 230-nm-thick thermally grown SiO2 dielectric layer (with an n-doping level of ~$3 \times 10^{17}$ cm$^{-3}$; capacitance 15 nF/cm$^2$).

**Device fabrication**. Solvent-induced precipitation growth of PTCDI-C8 nanowires was attained by injecting 16.7 ml of a PTCDI-C8 solution in chloroform (concentration = 0.25 mg/ml) into 250 ml of a bad solvent, i.e., ethanol. The supramolecular nanowires formation occurs within seconds. Subsequently, the PTCDI-C8 nanowires network was fabricated by a vacuum-assisted stamp method. This deposition method consists of vacuum filtration of PTCDI-C8 nanowire ethanol suspension through PTFE filters (Omnipore, 0.45 μm PTFE membrane) and then stamping it on a substrate and squeezed by a PTFE block.

After a few minutes, the dried Omnipore filter was peeled off and leaving the PTCDI-C8 nanowires network adhered onto different substrates, such as quartz or silicon substrate. The thickness of the PTCDI-C8 nanowires network can be controlled by adjusting the volume of added suspension solution during the filtration process, in our experiment the thickness of vertical transistors is in the range of 500 nm to 2.9 μm.

The Au nanomesh electrode on top of the 230 nm layer of SiO2 with 80 nm Al was patterned by nanosphere lithography[48,49]. First, a 10 wt% aqueous suspension of 800 μm diameter polystyrene (PS) nanospheres was diluted with an equal volume of anhydrous ethanol. Then, a clean glass slide was placed in a Petri dish filled with deionized water. Droplets of the PS nanosphere suspension were dropped onto the glass slide with a pipette and the nanospheres self-assembled into a monolayer consisting of small domains on the water surface. To obtain close-packed microsphere monolayer, 1 wt% sodium dodecyl sulfate was used as surfactant. The patterned nanospheres were then transferred to solid substrates by simply scooping the monolayer with the substrate. The monolayer is dried in air at room temperature. Eighty nanometer of gold nanomesh scaffold was then removed from the substrate by using FeCl3 as the etchant to dissolve the Al sacrifice layer. After washing several times with water, the floating nanomesh scaffold can be transferred to the target substrate by using the water-assistance transfer approach[50].

For the fabrication of PTCDI-C8 nanowires network-based vertical field-effect transistors and phototransistors, we used CVD graphene supported on Cu foil which was kindly provided by LG electronics Inc. The monolayer graphene was transferred onto SiO2/Si substrate by using PMMA-assistant transfer method[51]. The monolayer graphene was patterned through the photolithography and oxygen plasma, then deposited 40 nm gold electrodes for better contact. Following, the PTCDI-C8 nanowires network was transferred onto the graphene electrode by the vacuum-assisted stamp method mentioned above. At last, gold nanomesh scaffold was transferred onto the PTCDI-C8 nanowires network as the top drain electrodes through the water-assistance transfer approach. The use of water-assistance transfer approach in vertical organic electronic devices represents an effective solution for overcoming the typical difficulty associated with the deposition of top-contact metal electrodes by traditional methods and solves also any problem due to the permeation of metallic electrodes through the porous supramolecular network. Hence, this soft contacting method is demonstrated to be an effective protocol for the formation of metal/organic contact. Although water is involved during device fabrication, we should note that the moisture does not affect the electrical properties of the hierarchically assembled PTCDI-C8 supramolecular nanowires. The active area of VFETs is defined by the overlapping area between graphene and the top drain electrode (nanomesh scaffold), which is ca. 1.13 mm$^2$.

For the PPG measurement in the air, CYTOP thin film was used as the encapsulation layer by spin-coating CYTOP solution (CTL-809M and CT-Sol.180; ratio 1:1) at 2000 rpm, 50 s on the PTCDI-C8 nanowires VPTs then annealed at 90 °C, 30 min in the glovebox.

**Optoelectronic measurements**. All of the optoelectronic characterizations were performed inside a nitrogen-filled glovebox. For the irradiation, we used Polychrome V as a monochromatic source (300–690 nm), which was purchased from Till Photonics. For the NIR light measurement, we used 740 nm LED (Thorlabs, M740F2), 850 nm LED (OSLON, ILH-IO04-85NL-SC201-WIR200) and 940 nm LED (OSLON, ILH-IO04-94SL-SC201-WIR200) as the light sources. These irradiance levels can be further reduced by introducing calibrated ND filters into the optical system. Three ND filters (Thorlabs) were used: NE10A (ND = 1.0), NE20A (ND = 2.0), and NE30A (ND = 3.0). Three filters featuring the neutral densities (ND) of 1.0, 2.0 and 3.0 were used to reduce incident irradiance approximately by a factor of $10^1$, $10^2$, and $10^3$, respectively. The lowest accessible irradiance was ~10 pW/cm$^2$. The output light power has been calibrated by The PM100A Power Meter from Thorlabs. Coupled with the monochromatic light source, a Keithley 2636A system source meter was for the optoelectrical characterization.

For the noise measurement, the device was placed in a black metal box and the noise current of the device was measured using a phase-locked amplifier (Stanford Research Systems, SR830) at a bias voltage of 5 V. Before the noise measurements, the noise limit of the instrument was determined.

Photoplethysmography (PPG) test was performed in air with the 530 nm green light (Thorlabs, M505F3) as the light source. In the experiment, the output current of VPTs was monitored by a Keithley 2635 system source meter and customized Labview software.

**Computational details**. All electronic structure calculations were carried out via density functional theory (DFT) and time-dependent DFT (TDDFT) with PBE0 functional and 6-31 G(d) basis set in quantum chemistry package Q-Chem 5.2[52]. Polarizable continuum model (PCM) was applied to address the solvent effect in solution[53]. QM/MM method was applied to compute the electronic structure of the PTCDI-C8 nanowire[28]. Transition characters of the excited states were analyzed via the attachment–detachment densities and the Boys localized diabatization method. The absorption spectrum of PTCDI-C8 in solution was computed via thermal vibrational correlation function (TVCF) method in molecular material prediction package MOMAP 2019B[54].

## Data availability

The data that support the plots within this paper and other findings of this study are available from the corresponding author upon reasonable request.

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

## Acknowledgements

This work was financially supported by EC through the ERC project SUPRA2DMAT (GA-833707) and the Graphene Flagship Core 3 project (GA- 881603), the Labex projects CSC (ANR-10-LABX-0026 CSC) and NIE (ANR-11-LABX-0058 NIE) within the Investissement d'Avenir program ANR-10-IDEX-0002-02 and the International Center for Frontier Research in Chemistry (icFRC) and the Chinese Scholarship Council.

## Author contributions

Y.Y. and P.S. conceived the experiment and designed the study. Y.Y. performed the experiments and developed the fabrication method. Q.O. and Z.S. performed the theoretical calculation. K.W. and H.P. performed SEM characterization. H.P. carried out TGA measurement. Y.W. performed the Raman analysis. D.I. prepared nanowires networks. Y.Y. measured the optoelectronic devices. F.F and Y.S. did the noise current spectra measurements. Y.Y., Q.O., Z.S., and P.S. co-wrote the paper. All authors discussed the results and contributed to the interpretation of data as well as to the editing of the manuscript.

## Competing interests

The authors declare no competing interests.
