## [Peer Review File · Nature Communications]

Reviewer #1 (Remarks to the Author):

The authors report a sensitive NIR photodetector using PCDI nanowires and vertical FET structure. The assembly of PCDI resulted in the appearance of absorptions in the long-wavelength region. The device structure is unique, which is favorable for improving the photodetecting performance. The device performance is remarkable. I recommend its publication after the following revisions.

1. The discussions of supramolecular assembly of PCDI molecules needed to be strengthened in the revised manuscript. They did not discuss the ways to tune the lengths and widths of nanowires. In particular, the dependence of tail absorptions of nanowires on the preparation conditions was not discussed.
2. The interfacial structures between the nanowires and graphene and gold electrodes are crucial for the device performance. But, the characterizations of the interfacial structures were not well performed.
3. It is also important to show the reproducibility of the device performance.

Reviewer #2 (Remarks to the Author):

This work deals with organic photodetectors. The authors show that in contrast with the common view, it is possible to obtain devices displaying efficient response in the near-infrared (NIR) using wide-bandgap semiconducting materials. Thus, the authors were able to construct an efficient phototransistor based on self-assembled supramolecular nanowires of PTCDI-C8, a perylene derivative. The authors found that devices based on PTCDI-C8 nanowires show a strong photo-response when illuminated with the 850 nm light. However, such response was not observed in devices based on the thermal evaporated PTCDI-C8 films.

The experimental part is interesting. However, the discussion lacks clarity.

- (i) Why only the results for 850 nm light illumination are shown? Do the devices display any response at let's say 750 nm or 950 nm light illumination?
- (ii) It is difficult to compare the absorption spectra of thermal evaporated PTCDI-C8 film and PTCDI-C8 nanowires because the results are given in arbitrary units.
- (iii) The major drawback of this work is the lack of explanation of the origin of NIR absorption tail in nanowires.
- (iv) The DFT calculations show that the intermolecular charge-transfer energies are slightly smaller than the energy of the lowest intramolecular excited state. But these CT energies (2.1 eV) are still much larger than the 850 nm (1.5 eV) absorption. Why have then the authors concluded that intermolecular CT states determine the photo-response at 850 nm?
- (v) In the first sentence, the authors mentioned the lack of organic photodetectors capable to respond to illuminations at wavelengths exceeding 780 nm. Where 780 value is coming from?
- (vi) Last sentence in the introduction is incomplete.

Reviewer #3 (Remarks to the Author):

The paper by Yao and co-workers provide a new idea for near-infrared light (NIR) detection in organic optoelectronics, i.e., take the advantage of supramolecular self-assembly to trigger the emergence of efficient intramolecular charge transfer. The novel vertical phototransistors combined the 3D network of supramolecular nanowires with gold nanomesh scaffold exhibit fast photoresponse (~ 10 ms) and ultra-high sensitivity to monochromatic light from visible to NIR range, with photoresponsivity of 2×10^5 A/W and 1×10^2 A/W, at 570 nm and 940 nm, respectively, which the performance is even better than some narrow-bandgap polymers. More importantly, authors have demonstrated the practical application of supramolecular nanostructures in photoplethysmogram measurement, providing clear evidence of the huge potential of the supramolecular-based photodetectors for practical implementation as a highly efficient photodetection platform.

These are impressive figures of merit that further render this work of high impact. The paper results are undoubtedly interesting and the photo-electrical characterizations are performed meticulously; the explanations provided throughout the text of the manuscript are sound and consistent with the experimental results. The manuscript is well written in a clear style and appropriately illustrated, easy to read for a broad audience. I, therefore, recommend its acceptance once the minor points below are addressed:

- (1) In Fig. 3b, the authors only mentioned the light irradiance change from 0 to 100%, but what's the value of irradiance (mW/cm^2) at 100% intensity?
- (2) Why authors use 530 nm green light to perform the photoplethysmogram measurement instead of infrared light since your devices demonstrate very good IR response?
- (3) In supporting information, some Celsius symbols are displayed incorrectly, please carefully check and revise them.

Point-by-point reply

Reviewer #1

General comment_R1: *The authors report a sensitive NIR photodetector using PCDI nanowires and vertical FET structure. The assembly of PCDI resulted in the appearance of absorptions in the long-wavelength region. The device structure is unique, which is favorable for improving the photodetecting performance. The device performance is remarkable. I recommend its publication after the following revisions.*

Reply to the general comment_R1: We thank the Reviewer 1 for the positive assessment of our work. We have considered all the comments and made the requested revisions accordingly.

Comment_R1.1: *The discussions of supramolecular assembly of PCDI molecules needed to be strengthened in the revised manuscript. They did not discuss the ways to tune the lengths and widths of nanowires. In particular, the dependence of tail absorptions of nanowires on the preparation conditions was not discussed.*

Author reply R1.1: We thank the reviewer for asking more insight into the assembly process to form the nanowires and networks thereof, as well as their controllable optical properties. We totally agree with the reviewer that the quality of nanowires has a strong influence on their absorption spectrum. In our previous works, we have studied in detail the effect of different solvents and preparation methods on the size (cross section and length) as well as crystallinity of the nanowires (Nat. Nanotechnol., 2016, 11, 900-906; Adv. Mater. 2017, 1605760). In order to prepare a uniform network of highly crystalline and interconnected nanowires, the flexibility of nanowires is crucial, hence we used the solvent-induced precipitation (SIP) method with an ad hoc non-solvent (i.e. ethanol). To better illustrate the controlled supramolecular assembly process of PTCDI-C8 molecules forming the nanowire networks, we have provided more information in the main text.

Our action R1.1: In the main text at page 4, we have changed “N,N'-dioctyl-3,4,9,10-perylenedicarboximide (PTCDI-C8) molecules were self-assembled into supramolecular nanowires by solvent-induced-precipitation...” to “High quality N,N'-dioctyl-3,4,9,10-perylenedicarboximide (PTCDI-C8) n-type crystalline nanowires can be self-assembled by making use of either solvent phase-transfer (PT) or solvent-induced precipitation (SIP). Both these two processing strategies enable to achieve a dynamic balance between the orthogonal good and poor solvent. Significantly, the properties of the formed nanowires including the degree of crystallinity, size and softness could be further tuned by modifying concentration, temperature, and solvent type [ref. 26]. PTCDI-C8 nanowires prepared by phase transfer method are more crystalline and exhibit red shifted resonance absorption peaks, however the high rigidity and large size (both in length and cross section) of PT-nanowires make them not ideal for the generation of homogeneous supramolecular films [ref. 27]. Because of these downsides we decided to exploit solvent-induced-precipitation to

self-assemble the PTCDI-C8 molecules into supramolecular nanowires. Such a crystal growth process triggered by the addition of a bad solvent (i.e. ethanol) minimizes the incorporation or trapping of solvent molecules into the crystals, being an optimal strategy to achieve enhanced opto-electronic properties of the nanostructures [ref. 22]. In addition, PTCDI-C8 nanowires prepared by SIP exhibit a high flexibility which render them most suitable for the preparation of a uniform closely-packed 3D nanowires network.”.

Comment_R1.2: *The interfacial structures between the nanowires and graphene and gold electrodes are crucial for the device performance. But, the characterizations of the interfacial structures were not well performed.*

Author reply R1.2: We thank the reviewer for giving us the chance to clarify this point. We agree with you that the interface between nanowires, graphene/SiO₂ and gold nanomesh are very important in order to control various physical processes occurring at interfaces. To better demonstrate the quality of the interfacial contacts, high-resolution cross-sectional SEM images have been added in the Supplementary Information as Figure S11.

Our action R1.2: In the Supplementary Information, we have added Figure S11 d-f, on page S8. Moreover, in the main text, we have changed “...revealing a stacked architecture with well-defined interfaces between each component without any collapsing, especially between nanowires network and the nanomesh scaffold.” into “...revealing a stacked architecture with well-defined and sharp interfaces between each layer without any collapsing, especially between the nanowires network and the nanomesh scaffold.”, on page 8.

Comment_R1.3: *It is also important to show the reproducibility of the device performance.*

Author reply R1.3: Thank you for your kind suggestion. Our devices displayed very good reproducibility and stability as evidenced in the additional data which were integrated in the Supplementary Information at Figure S23 and S24.

Our action R1.3: In Supplementary Information, we have added Figure S23 and S24 to provide evidence for the good repeatability and stability of our devices. In main text at page 13, we have added “Our devices also demonstrate good reproducibility and stability (see Supplementary Figure 23 and 24).“

Reviewer #2

General comment_R2: *This work deals with organic photodetectors. The authors show that in contrast with the common view, it is possible to obtain devices displaying efficient response in the near-infrared (NIR) using wide-bandgap semiconducting materials. Thus, the authors were able to construct an efficient phototransistor based on self-assembled supramolecular nanowires of PTCDI-C8, a perylene derivative. The authors found that*

devices based on PTCDI-C8 nanowires show a strong photo-response when illuminated with the 850 nm light. However, such response was not observed in devices based on the thermal evaporated PTCDI-C8 films. The experimental part is interesting. However, the discussion lacks clarity.

Reply to the general comment_R2: We would like to thank Reviewer 2 for the positive evaluation of our work. We also appreciate all the constructive comments and suggestions which we believe are instrumental to improve the quality of our work. We have considered all the points raised and made the requested revisions accordingly.

Comment_R2.1: *Why only the results for 850 nm light illumination are shown? Do the devices display any response at let's say 750 nm or 950 nm light illumination?*

Author reply R2.1: We fully agree with the Reviewers about the importance to show the response of the device to other wavelengths. To better demonstrate the NIR light response of thermally evaporated PTCDI-C8 film and nanowires networks, we have added the results of their photoresponse at 740 nm and 940 nm.

Our action R2.1: In main text at page 5, we have changed “Supplementary Figure 4 showed a striking photoresponse to 850 nm NIR light” to “Supplementary Figure 4 showed a striking photoresponse to 740 nm, 850 nm and 940 nm NIR light”. Moreover, in the Supplementary Information, we have added Figure S4 e, f and Figure S5 e, f, on page S4 and S5 displaying the transfer and output characteristics for networks and thermally evaporated films when irradiated with a 740 nm and 940 nm light (with channel length of 10 μm).

Comment_R2.2: *It is difficult to compare the absorption spectra of thermal evaporated PTCDI-C8 film and PTCDI-C8 nanowires because the results are given in arbitrary units.*

Author reply R2.2: We thank the reviewer for bringing up this point. We have updated the absorption spectra of thermal evaporated PTCDI-C8 film and PTCDI-C8 nanowires with the actual absorbance value shown on the Y-axis.

Our action R2.2: In the Supplementary Information, we have updated Figure S3 a, on page S3 now displaying the actual absorbance in the Y-axis.

Comment_R2.3: *The major drawback of this work is the lack of explanation of the origin of NIR absorption tail in nanowires.*

Author reply R2.3: The origin of the NIR absorption tail can be understood theoretically via the transition character of the S1 state in PTCDI-C8 dimer as depicted in Figure 1. The engagement of large portion of charge-transfer character in S1 gives rise to more significant geometrical adjustment upon excitation and hence larger reorganization energy between S1 and the ground state. Because of the increase in the geometrical change and the reorganization energy, the absorption spectrum of the aggregates will be broadened compared to that in

solution, of which the S1 state has pure local-excitation character. Theoretical demonstration of the reorganization energy induced broadening of the spectra can be found in Acta Phys. Chim. Sin., 2010, 26(4): 1051-1058.

Our action R2.3: In the main text at page 6, we have added the following paragraph: “We found that intermolecular charge-transfer (CT) transition between two monomers (CT1 and CT2 in Fig.1) are dominant with total contributions exceeding 80%, yielding a decrease of the energy of S1 compared to the value in solution (Supplementary Table 1) along with a significant geometrical adjustment upon excitation with larger reorganization energy between S1 and the ground state. Such geometrical change and enlarged reorganization energy significantly broaden the absorption spectrum [ref. 30], resulting in an elongated tailing peak in the absorption spectrum at the NIR region (Supplementary Figure 3 and 6). The absorption tail induced photoresponse of PTCDI-C8 can also be evinced by the fact that the response from 570 nm to 940 nm is monotonically weakened as shown in Supplementary Figure 22, which is in good agreement with the monotonic decrease of the photon absorption at wavelength exceeding 570 nm. Altogether, such findings confirms that intermolecular CT determines the photoresponse of PTCDI-C8 nanowires in the NIR region.”.

Comment R2.4: *The DFT calculations show that the intermolecular charge-transfer energies are slightly smaller than the energy of the lowest intramolecular excited state. But these CT energies (2.1 eV) are still much larger than the 850 nm (1.5 eV) absorption. Why have then the authors concluded that intermolecular CT states determine the photo-response at 850 nm?*

Author reply R2.4: It should be noted that the response at 850 nm of PTCDI-C8 is not necessarily the origin of the S1 excitation. As we have discussed in “our action R2.3” above, the response at the NIR region is mainly due to the absorption tail that results from the charge-transfer character of the S1 state in aggregates, while the excitation energy of S1 (2.1 eV) should actually correspond to the absorption maximum. This can be evinced by the fact that the response from 570 nm to 940 nm is monotonically weakened as shown in Supplementary Figure S22. The photoresponse reaches its maximum at 570 nm, which is in good agreement with the excitation energy of the CT state (2.1 eV).

Our action R2.4: We have discussed this point in thoroughly at page 6 of the main text as detailed above in our action R2.3.

Comment R2.5: *In the first sentence, the authors mentioned the lack of organic photodetectors capable to respond to illuminations at wavelengths exceeding 780 nm. Where 780 value is coming from?*

Author reply R2.5: The first sentence states: “Near-infrared organic photodetectors (OPDs), capable to respond to illuminations at wavelengths exceeding 780 nm, have attracted a notable interest during the last few years because of their potential application as motion detector, remote control, night vision, biomedical imaging as well as health monitoring.”.

Indeed at 780 nm the infrared region starts. Hence, the sentence above refers to the near-infrared region of the electromagnetic spectrum (Chem. Soc. Rev., 2020,49, 653-670).

Comment_R2.6: *Last sentence in the introduction is incomplete.*

Author reply R2.6: We thank the reviewer for bringing up this point to our attention. We have made corresponding change to the last sentence of the introduction.

Our action R2.6: In the main text at page 4 we have added “generation highly sensitive NIR photodetectors” after “and therefore in may represent a major step forward for the next”.

Reviewer #3

General comment_R1: *The paper by Yao and co-workers provide a new idea for near-infrared light (NIR) detection in organic optoelectronics, i.e., take the advantage of supramolecular self-assembly to trigger the emergence of efficient intramolecular charge transfer. The novel vertical phototransistors combined the 3D network of supramolecular nanowires with gold nanomesh scaffold exhibit fast photoresponse (~10 ms) and ultra-high sensitivity to monochromatic light from visible to NIR range, with photoresponsivity of 2×10^5 A/W and 1×10^2 A/W, at 570 nm and 940 nm, respectively, which the performance is even better than some narrow-bandgap polymers. More importantly, authors have demonstrated the practical application of supramolecular nanostructures in photoplethysmogram measurement, providing clear evidence of the huge potential of the supramolecular-based photodetectors for practical implementation as a highly efficient photodetection platform.*

These are impressive figures of merit that further render this work of high impact. The paper results are undoubtedly interesting and the photo-electrical characterizations are performed meticulously; the explanations provided throughout the text of the manuscript are sound and consistent with the experimental results. The manuscript is well written in a clear style and appropriately illustrated, easy to read for a broad audience. I, therefore, recommend its acceptance once the minor points below are addressed:

Reply to the general comment_R1: We would like to thank Reviewer 3 for taking the time to evaluate our article, and for the very positive assessment. We have considered all the comments and made the requested revisions accordingly.

Comment_R3.1: *In Fig. 3b, the authors only mentioned the light irradiance change from 0 to 100%, but what's the value of irradiance (mW/cm²) at 100% intensity?*

Author reply R3.1: We thank the reviewer for pointing this out. To address this question, we have added light power density spectrum of the monochromatic light source (100% intensity) in Figure S16 of the Supplementary Information.

Our action R3.1: In Supplementary Information at page S11, we have added Figure S16c to demonstrate the light irradiance of monochromatic light source. In the main text on page 10, we have added “Figure S16c” into “...its light absorption property (Supplementary Figure 6)”.

Comment R3.2: *Why authors use 530 nm green light to perform the photoplethysmogram measurement instead of infrared light since your devices demonstrate very good IR response?*

Author reply R3.2: Thank you for your kind suggestion. Although our device demonstrated notably high NIR light response, the photoresponsivity of VPTs in the NIR region is still much lower than in the visible region (3.1×10^2 A/W@740 nm vs 1.7×10^5 A/W@570 nm). As shown below, when 660 nm red light is used, no clear PPG signal can be detected.

Comment R3.3: *In supporting information, some Celsius symbols are displayed incorrectly, please carefully check and revise them.*

Author reply R3.3: We have corrected the Celsius symbols and marked them with yellow background in the Supplementary Information at page S10 and S11, in Figure S15 and Table S2.

Reviewer #1 (Remarks to the Author):

The authors have well addressed my concerns and appropriately incorporated the respective corrections in the revised manuscript. Thus, I recommend its publication.

Reviewer #2 (Remarks to the Author):

The revised manuscript is more clear and addressed all reviewers' comments and suggestions. I recommend publication.

Reviewer #3 (Remarks to the Author):

The authors have addressed the reviewer's concerns, and the reviewer suggested the acceptance of the manuscript.

Point-by-point reply

Reviewer #1

General comment_R1: *The authors have wel addressed my concerns and appropriately incorporated the respective corrections in the revised manuscript. Thus, I recommend its publication.*

Reply to the general comment_R1: We are grateful to the reviewer 1 for supporting the publication of this work in Nature Communications.

Reviewer #2

General comment_R2: *The revised manuscript is more clear and addressed all reviewers' comments and suggestions. I recommend publication.*

Reply to the general comment_R2: We would like to thank the reviewer 2 for supporting the publication of this work in Nature Communications.

Again, we sincerely appreciate all the referee for his/her valuable comments and suggestions that greatly helped us to improve the quality of this article.

Reviewer #3

General comment_R3: *The authors have addressed the reviewer's concerns, and the reviewer suggested the acceptance of the manuscript.*

Reply to the general comment_R3: We are really appreciating the reviewer 3 for recommending the publication of this work in Nature Communications.